# Research Progress on the Regulation of Autophagy and Apoptosis in Insects by Sterol Hormone 20-Hydroxyecdysone

**DOI:** 10.3390/insects14110871

**Published:** 2023-11-12

**Authors:** Luobin Lin, Huaqin Li, Qinzhou Zheng, Jiaxuan Hu, Wenmei Wu

**Affiliations:** 1Guangdong Province Key Laboratory of Biotechnology Drug Candidates, School of Life Sciences and Biopharmaceuticals, Guangdong Pharmaceutical University, Guangzhou 510006, China; 2100920428@gdpu.edu.cn (L.L.); 2100920461@gdpu.edu.cn (Q.Z.); 2School of Health Sciences, Guangzhou Xinhua University, 19 Huamei Road, Tianhe District, Guangzhou 510520, China; huaqinli1118@163.com; 3State Key Laboratory of Biocontrol, Guangdong Key Laboratory of Pharmaceutical Functional Genes, School of Life Sciences, Sun Yat-sen University, Guangzhou 510275, China; hujiaxuan99@163.com

**Keywords:** 20-Hydroxyecdysone, insect physiology, autophagy, apoptosis, pest control strategies

## Abstract

**Simple Summary:**

Insects rely on a sterol hormone known as 20-Hydroxyecdysone to regulate vital cellular activities. This study delves deep into the biosynthesis of this hormone, its signal transduction, and the mechanisms by which it regulates autophagy (cellular self-cleaning) and apoptosis (programmed cell death). Our primary goal was to unravel the intricate interplay between 20-Hydroxyecdysone and these cellular processes. Through our research, we uncovered the intricate pathways and mechanisms that this hormone influences in cell renewal and controlled cell death. These insights not only hold significance for comprehending insect development and survival but also have broader implications in fields like agriculture and pest control. By shedding light on these processes, we aim to provide valuable knowledge that can be leveraged for the betterment of our environment and society.

**Abstract:**

20E (20-Hydroxyecdysone) is a central steroid hormone that orchestrates developmental changes and metamorphosis in arthropods. While its molecular mechanisms have been recognized for some time, detailed elucidation has primarily emerged in the past decade. PCD (Programmed cell death), including apoptosis, necrosis, efferocytosis, pyroptosis, ferroptosis, and autophagy, plays a crucial role in regulated cell elimination, which is vital for cells’ development and tissue homeostasis. This review summarizes recent findings on 20E signaling regulated autophagy and apoptosis in insects, including *Drosophila melanogaster*, *Bombyx mori*, *Helicoverpa armigera*, and other species. Firstly, we comprehensively explore the biosynthesis of the sterol hormone 20E and its subsequent signal transduction in various species. Then, we focus on the involvement of 20E in regulating autophagy and apoptosis, elucidating its roles in both developmental contexts and bacterial infection scenarios. Furthermore, our discussion unfolds as a panoramic exposition, where we delve into the fundamental questions with our findings, anchoring them within the grander scheme of our study in insects. Deepening the understanding of 20E-autophagy/apoptosis axis not only underscores the intricate tapestry of endocrine networks, but also offers fresh perspectives on the adaptive mechanisms that have evolved in the face of environmental challenges.

## 1. Introduction

Ecdysteroids, acknowledged as the principal steroid hormones in arthropods, are commonly known as “molting hormones” in insect physiology [1]. Ecdysone serves as a prohormone crucial for synthesizing the primary insect molting hormone, 20E. Once ecdysone is released, it undergoes conversion into the more active form, 20E. This hormone is renowned for its central role in regulating various life stages in insects, facilitating the smooth transition from larval stages to pupation and eventually to maturity. It exerts a significant influence on cellular processes such as autophagy and apoptosis, which are essential for the developmental changes that occur during molting [2]. These cellular events are integral to insect development, growth, and survival, especially in species like *D. melanogaster* (*Drosophila melanogaster*), *B. mori* (*Bombyx mori*), *H. armigera (Helicoverpa armigera*), and others [3,4,5,6].

PCD is a vital cellular mechanism in insects, ensuring the proper development, maintenance, and survival [7,8]. Processes such as apoptosis, necrosis, and autophagy play pivotal roles in tissue remodeling and the removal of damaged cells [9]. In advanced eukaryotes, three main autophagy types have been identified: macroautophagy, microautophagy, and chaperone-mediated autophagy. Both microautophagy and chaperone-mediated autophagy direct cytosol segments and proteins to lysosomes, either autonomously or with assistance by chaperones [10]. Macroautophagy, often referred to as autophagy, is a recycling mechanism crucial for cellular homeostasis, guiding developmental shifts and addressing environmental stresses, especially in insects [11]. Apoptosis emerges as a pivotal process-shaping development, homeostasis, and defense [12]. Autophagy and apoptosis in insects have been extensively studied, not only in the midgut epithelium of *Acheta domesticus* (*Insecta*, *Orthoptera*, and *Gryllidae*), but also in the context of insect infections [13,14]. This indicates that both autophagy and apoptosis play indispensable roles in cellular development. Furthermore, 20E emerges as a key regulator of autophagy and apoptosis. In the silk-producing *B. mori*, a lepidopteran species, 20E’s regulation of autophagy and apoptosis are vital for its unique life cycle stages [15]. Similarly, the malaria vector, *Anopheles gambiae*, as an *Diptera* species depends on the processes regulated by 20E for its rapid larval development [16]. On the other hand, in the cotton bollworm, *H. armigera*, the interplay between 20E and other signaling pathways modulates PCD, which is crucial for its adaptability to diverse environments [6]. Collectively, 20E regulates autophagy, apoptosis, immune function, and the central nervous system, indicating high antioxidant activities and other functions [17]. This evolutionary pattern highlights the unique biological responses across different life phases. Therefore, it is crucial to assess regulatory mechanisms through the lens of age and developmental transitions.

The understanding between 20E, autophagy, and apoptosis holds profound implications beyond basic biology. In agriculture, a comprehensive understanding of this triad could pave the way for innovative pest control methods, reducing the reliance on environmentally harmful pesticides. This synergistic connection between 20E and autophagy/apoptosis in insects not only underscores the complexity of nature but also presents a fertile ground for potential advancements in both sustainable agriculture and medicine.

## 2. Sterol Hormone 20E Biosynthesis

From a biochemical perspective, the sterol hormone 20E, which belongs to the ecdysteroid family, plays a pivotal role in orchestrating insect growth, development, and reproduction [18,19]. Its biosynthesis begins with the assimilation of dietary cholesterol, serving as the foundational precursor. The process of 20E biosynthesis in insects holds a central process governing the development and metamorphosis [20]. Biosynthesis of 20E involves a series of enzymatic reactions that convert cholesterol into the active hormone. Initially, cholesterol undergoes a hydroxylation process, which is catalyzed by several cytochrome P450 enzymes, leading to the production of precursors such as ecdysone. Subsequent enzymatic steps, including further hydroxylation and oxidation, result in the formation of 20E. Upon synthesis, 20E targets specific nuclear receptors, mainly the *EcR* (ecdysone receptor), to regulate gene expression essential for developmental processes [21]. This bioactive form of ecdysone then participates in the regulation of gene expression by forming a heterodimer with USP (ultraspiracle protein), which is essential for initiating the developmental processes [22]. The genes *EcR-B1* (ecdysone receptor B1 isoform) and *USP1* (ubiquitin specific peptidase 1) have been predominantly associated with 20E regulation [23,24]. Intriguingly, other pivotal factors such as *E75B* and *Ha-eIF5c* have also been identified to influence 20E regulation. It was elucidated that the genes modulated by 20E, namely *EcR-B1*, *USP1*, *E75B*, *BR-CZ2*, *HHR3*, and *Ha-eIF5c*, are influenced by the suppressive actions of both *Ha-Ntf2* and *Ha-Ran* [25,26,27]. Taken together, these evidences suggest that the regulation of 20E exhibits conserved mechanisms across diverse species. These information provide a coherent framework, tracing from its formation, through its process, and to its ultimate effects. Therefore, a comprehensive understanding of 20E biosynthesis yields profound insights into its subsequent roles in biological metabolism.

E (Ecdysone), which is the immediate precursor of 20E, is synthesized and released via specialized structures in insects, such as the prothoracic glands or ovaries. After secretion, it circulates in the hemolymph and is transformed into the biologically active 20E in peripheral tissues. This conversion primarily occurs in the fat body, midgut, and malpighian tubules, particularly during the larval stage [28]. Illustratively, in *Drosophila* and other members of the *Diptera* order, ecdysone is synthesized and secreted via the prothoracic glands, acting as the seminal trigger for the initiation of the molting cascade [29]. Subsequently, E undergoes hydroxylated at the 20th position by the enzyme, shade, leading to the formation of the biologically activated 20E being released from the fat body and transported to different cells and tissues as needed. In the adults of some insects, the presence of ecdysteroids in the ovary was discovered 40 years ago, and now it is well established that ovarian follicular cells synthesize ecdysone de novo [30,31]. Intriguingly, in certain lepidopteran species, it has been documented that male gonads release significant quantities of ecdysone in vitro [32].

The biosynthetic cascade is triggered when the PTTH (neuropeptide Prothoracicotropic Hormone) binds to its receptor, *Torso*, on prothoracic gland cells [33]. This interaction initiates the Ras/Raf/ERK signaling pathway, activating genes crucial for ecdysone biosynthesis [34]. Simultaneously, the Insulin/IIS (IGF Signaling) pathway, responsive to nutritional cues, enhances ecdysone synthesis by increasing the expression of biosynthetic enzymes and by augmenting the prothoracic glands’ sensitivity to PTTH [35]. The biosynthetic process is further modulated by biogenic amines, particularly octopamine, and neuropeptides such as *B. mori*, although their roles may vary across different species. Significantly, external factors such as photoperiodicity and ambient temperature subtly influence this biosynthetic pathway [36,37,38,39]. This biosynthesis involves cytochrome P450 enzymes, with six identified as playing roles in the 20E pathway [40]. While the cytochrome P450 Cyp6t3 has been shown not to be essential for ecdysone biosynthesis in *D. melanogaster*, CYP18A1 is still crucial for its metamorphosis [41,42]. The biosynthesis of 20E involves multiple cytochrome P450 enzymes, including CYP306A1, which facilitates the production of the arthropod molting hormone [43]. These findings provide a new perspective, suggesting that P450 may have a more pronounced role in regulating the sterol hormone related to cellular processes.

Moreover, JH (Juvenile Hormone) adds an additional layer of regulation, often acting antagonistically to 20E. Elevated JH concentrations typically suppress 20E synthesis [44]. In the captivating world of beetles, as part of the *Coleoptera* order, the JH takes center stage, driving vitellogenesis within the fat body [45,46]. Vitellogenesis involves the synthesis and accumulation of yolk proteins in maturing oocytes, which are crucial for nourishing the embryo post-fertilization [47]. This crucial process primarily occurs in the fat body, an organ comparable to the mammalian liver. Interestingly, in the adult females of *Hemimetabola* (*Dictyoptera* to *Hemiptera*) and *Coleoptera*, JH serves as the main regulator, exerting pleiotropic controls over various aspects of female reproduction [48,49]. Given the diversity of *Coleoptera*, there is variability in how different beetle species respond to JH, reflecting their specific reproductive strategies and ecological niches. Furthermore, the coordinated interplay of these PCD forms, observed in the ovarian tropharia of certain beetles, underscores the delicate equilibrium insects establish to ensure their survival and adaptability amidst constantly evolving environments [50]. While ecdysone, 20E, and JH each govern insect growth and reproduction, an intriguing question arises: Could 20E also have a pivotal role in the insect reproductive processes? If so, what are the underlying mechanisms through which these hormones operate? Unraveling these questions could provide fresh insights into the multifaceted roles of steroid hormones in insect biology.

In the silkworm *B. mori*, the enzymatic activity of ecdysteroid-phosphate phosphatase plays a crucial role in the 20E biosynthetic pathway within eggs, catalyzing the conversion of maternal-conjugated ecdysteroids [5]. This process is paralleled in the species *Drosophila*, belonging to *Diptera*, where studies have uncovered a reciprocal regulatory relationship between 20E and JH in larvae [51]. Each hormone intricately influences the synthesis of the other, showcasing the complexity of hormonal interactions during development. Recent studies have revealed that 20E, which is specifically derived from cholesterol via the catalytic activity of a series of cytochrome P450 enzymes that hydroxylate E at carbon 20, particularly CYP314a1, is essential for insect molting and metamorphosis [52]. This underscores the significance of *Drosophila* E-20-monooxygenase, which is identified as the product of the shade locus. In a recent investigation focused on *H. armigera*, a representative of the *Lepidoptera* order, four cytochrome P450 homologs-HarmCYP302A1, HarmCYP306A1, HarmCYP314A1, and HarmCYP315A1, were identified. These homologs demonstrated evolutionary conservation across lepidopterans. Intriguingly, while HarmCYP302A1 and HarmCYP315A1 were predominantly expressed in the larval prothoracic glands, such consistent expression patterns were absent for HarmCYP306A1 and CYP314A1 [53]. It is evident that, at different times, the biological activity and expression of different enzymes can serve as indicators of biological activity to some extent. However, the question remains: Is the emergence of biological activity necessarily related to these enzymes?

In conclusion, the intricate process of 20E biosynthesis serves as a cornerstone in the field of insect physiology and development. A comprehensive grasp of this pathway not only sheds light into the evolutionary intricacies of insect life, but also offers promising opportunities for transformative applications in pest management, medicine, and biotechnology. As we unravel the steps and intricacies of 20E biosynthesis, we gain a deeper comprehension of insect biology and open avenues for sustainable and innovative solutions to challenges in agriculture and health.

## 3. 20E Signal Transduction

Specifically, 20E counteracts IIS through the AMPK (AMP-activated protein kinase) and PP2A (protein phosphatase 2A) axis within the insect fat body, leading to a suppression of growth rates. Increased levels of 20E activate AMPK, a pivotal molecular sensor responsible for maintaining energy equilibrium in the insect fat body, of which the subsequently activated AMPK stimulates PP2A, which in turn dephosphorylates the insulin receptor and AKT (protein kinase B), culminating in the inhibition of the IIS signaling pathway [15]. Moreover, *E93* (Ecdysone-induced protein 93), encoded by a member of the helix-turn-helix transcription factor family, also plays a crucial role in the crosstalk of 20E signaling with JH signaling, mediated by the JH primary response gene *Kr-h1* [54]. These transcription factor genes, described as “primary response genes”, are activated and transcribed within minutes after 20E stimulation, without requiring de novo protein synthesis. Upon binding to its receptor complex, EcR-USP, 20E rapidly and strongly induces the expression of primary-response genes. These genes, including nuclear receptors and transcription factors like *E75* (Ecdysone-induced protein 75), *E93*, *Br-C* (Broad-Complex), and *E74* (Ecdysone-induced protein 74), play vital roles in the larval–pupal metamorphosis of insects (Figure 1) [54,55,56].

The genes responsible for E biosynthesis are essential for producing the precursor hormone 20E, exhibiting distinct expression patterns in the tubular accessory glands of adult male insects [57]. In the remodeling fat body of *Drosophila*, which serves as an in vivo model, research has revealed a complex molecular interplay between autophagy and caspase activity. Despite initially seeming to be opposing processes, both autophagy and caspase activity are triggered via the common stimulus of 20E. Although they counterbalance each other, obstruction in one pathway amplifies the other. This dynamic interplay is further highlighted via the observed progressive augmentation of both autophagy and caspase activity in the remodeling fat body, triggered via a surge of the molting hormone, 20E, during the pivotal larval–prepupal transition [58]. Additionally, research has elucidated the antagonistic relationship between 20E and the insulin/IIS pathway.

In *B. mori*, a thorough analysis of the *EcR-A* and *EcR-B1* expression during the larval–pupal transition has revealed their upregulation by 20E, primarily in the lateral neurosecretory cells of the larval brain, which are known as prothoracicotropic hormone-producing cells [23]. Significantly, 20E signaling predominantly mediates the occurrence of autophagy in larval tissues and organs during larval molting and larval–pupal metamorphosis [59]. The 20E-EcR-USP complex initially induces the expression of a small set of primary response genes, including transcription factors such as *Br-C*, *E74*, *E75*, and *E93*, which are responsible for the upregulation of a larger set of downstream secondary response genes and crucial for successful molting or metamorphosis (Table 1). In a distinct study, it was demonstrated that 20E plays a role in the elongation of genital disks. The precursor hormone 20E significantly induced the expression of three different *BmE75* isofoms in *Bombyx*, which consequentially coordinated feedback to 20E biosynthesis [55]. Both BIGFLP (*Bombyx* imp-L2 gene-enhancer flippase protein) and 20E hormones were discovered to synergistically stimulate protein synthesis, where the modulatory effects were mediated via the IIS pathway and the mitogen-activated protein kinase pathway [60]. This coordination between BIGFLP and 20E underscores their joint regulation of genital disk growth and development. Notably, RNA interference have revealed that the downregulation of *BmADK* leads to the decreased expression of *ATG-8*, *Caspase-9*, *Ec-R*, *E74A*, and *Br-C* [61]. This suggests *BmADK*’s potential involvement in 20E signaling, apoptosis, and autophagy, all of which are essential for silk gland degeneration and silkworm metamorphosis. However, the precise role that 20E plays in these vital processes remains insufficiently elucidated.

Additionally, studies focusing on HaP60 (*H. armigera* P60) have revealed its phosphorylation under the regulation of insulin-like peptides during larval growth stages. In metamorphosis, the dephosphorylation of HaP60, coupled with its upregulation in response to 20E, orchestrates a series of protein phosphorylations, which leads to the cytosolic localization of forkhead box protein O subsequently promoting cellular proliferation, indicating a subtle connection that 20E has an impact on the additional regulation of autophagy and apoptosis [62]. Simultaneously, there was a significant downregulation of several 20E-responsive genes, such as *LdEcR*, *LdUSP*, *LdHR3*, and *LdFTZ-F1.* Intriguingly, the silencing of *LdTorso* also led to the upregulation of the JH biosynthesis gene larval diapause hormone acid methyltransferase, leading to elevated JH titers and the subsequent activation of the early-inducible JH gene, *LdKr-h1* [63].

Collectively, the cytochrome P450 enzymes and primary-response genes play a crucial role in the biosynthesis of 20E across different species (Table 1). The precise modulation of gene expression and cellular processes by 20E signaling underscores its evolutionary significance and its pivotal role in insect biology. Delving into the complexities of 20E signal transduction not only sheds light on the molecular mechanisms that drive insect development and metamorphosis, but also reveals potential strategies for targeted pest management and other practical applications.

**Table 1 insects-14-00871-t001:** The cytochrome P450 enzymes and primary-response genes in 20E signal transduction.

Species	Cytochrome P450 Enzymes	The Primary-Response Genes
*D. melanogaster*	DmCYP18a1 [41] DmCyp6a8/DmCyp6a2 [64] DmCyp6g1 [65] DmCYP302A1, DmCYP315A1 and DmCYP314A1 [66]	*DmE75A* [67] *DmE75B*, *DmE75C* and *DmE75D* [68] *DmE93*, *DmE74*, *Br-C*, and *β-FTZ-F1* [69]
*B. mori*	BmCyp450, BmCYP314a1, BmCYP315a1, BmCYP302a1 and BmCYP306a1 [70] BmCYP4 and CYP6 [71] BmCYP18A1 [72]	*BmE75A*, *BmE75B* and *BmE75C* [73] *BmE93* and *BR-C* [74] *EcR-A* and *BHR3* [75]
*H. armigera*	HarmCYP302A1, HarmCYP306A1, HarmCYP314A1, and HarmCYP315A1 [53] HarmCYP6b2, HarmCYP18a1 and HarmCYP18b1 [76]	*Br-C*, *E74*, *Ha-Ntf2*, *Ha-Ran*, *E75B*, *HHR3*, and *Ha-eIF5c* [25]

## 4. 20E Regulates Autophagy and Apoptosis in Insects

While the precise mechanisms underlying how 20E influences PCD are not yet fully elucidated, preliminary discoveries suggest that the potential involvement of the enzymatic machinery are responsible for 20E in the transformation of dietary sterols into ecdysteroids [4]. The steroid hormone 20E, with its diverse functions in regulating autophagy, apoptosis, and various cellular processes, is gradually emerging as a key player in the orchestration of insect growth and metamorphic transitions.

The study suggested that specific neurons, particularly *RP2s*, provide valuable insights into the molecular dynamics of neuronal demise during insect metamorphosis [77]. This process is likely regulated by steroid hormones, with 20E assuming a crucial role in coordinating both autophagy and apoptosis in these organisms. Moreover, it has been observed that 20E can influence intracellular calcium levels, consequently shifting from autophagic cell survival to apoptotic cell death [17,78]. Another research has illuminated the role of 20E in upregulating the transcription factor *Klf15* (Krüppel-like factor) by enhancing *Klf15* transcription in the metamorphic fat body through its nuclear receptor EcR, thereby promoting autophagy, apoptosis, and gluconeogenesis during metamorphosis [79]. Furthermore, studies have emphasized the reduction in glycolysis and *PGK1* expression levels during metamorphosis under the regulation of 20E. While insulin promotes glycolysis and cell proliferation via PGK1 phosphorylation, 20E counteracted this by dephosphorylating PGK1 via PTEN (phosphatase and tensin homolog), thereby inhibiting glycolysis [80]. Moreover, a separate study highlighted that even in the absence of E or 20E synthesis, specific mutants still exhibit transcriptional enhancement of glycolytic genes during embryogenesis [81]. This suggests that these hormones may not be crucial for this particular metabolic transition, further emphasizing the intricate role 20E plays in regulating PCD. During insect metamorphosis, cell death artfully transforms larval forms, paving the way for adulthood. Modern research into *Lepidoptera* and *Diptera* metamorphosis unveils the intricate ballet of tissue change and its governing regulations [82]. In the midgut, apoptotic cells are cast into the lumen, initially dispersed but eventually forming a cohesive layer beneath the peritrophic membrane. Intriguingly, only the youthful midgut epithelium shows apoptosis. As it matures, necrosis emerges alongside apoptosis, eventually taking precedence and ultimately overshadowing apoptosis completely [83]. Within the intricate cellular machinery of mammals, apoptosis is marked by the coordinated movement of Cyt-c (cytochrome C) from its protected location within the mitochondria to the expansive environment of the cytoplasm [84]. Understanding the roles of 20E can potentially inform pest control strategies, given the hormone’s influence on insect development and survival.

### 4.1. Regulation in D. melanogaster

In the developmental stages of *Drosophila*, the steroid hormone E plays a crucial role in guiding the transition from the embryo to the larva and subsequently to the pupa. It is notably during the larval–pupal metamorphosis that a significant wave of PCD occurs, efficiently eliminating redundant larval tissues. E’s periodic surges regulate the activation of distinct transcription factors, which, in turn, promote the expression of crucial genes associated with cell death [85]. This ensures that PCD is orchestrated both spatially and temporally. Furthermore, E governs cell death in specific tissues during both the larval and adult phases. Basically, a surge of 20E at the conclusion of the larval stage initiates metamorphosis. This process entails the removal of redundant larval tissues and the repurposing of molecular materials to form adult structures [30]. In the fat body, *E93* serves as a primary mediator for 20E signaling, directing the onset of both autophagy and caspase activity. This activation is triggered via a surge of the molting hormone, 20E, transmitted through the 20E nuclear receptor complex, EcR-USP [86]. In a separate study delving into the intricate process of *Drosophila* salivary gland degradation, researchers have elucidated that the transcriptional upregulation of mitochondrial apoptosis factors, including *Cyt-c* and genes encoding the death-associated APAF1-related killer, precedes the upregulation of genes encoding the initiator and effector caspases [87]. It was reported that the accumulation of transcripts from both *reaper* and *grim* is effectively curtailed by this steroid hormone, highlighting the intricate connection between 20E and the apoptotic process [88]. As the degradation progresses, there is a noticeable and gradual increase in the presence of the Cyt-c protein and active caspase 3 in the cytoplasm. Interestingly, the Cyt-c protein exhibits colocalization with mito-GFP, a definitive marker for cytoplasmic mitochondria. Concurrently, alternations in the mitochondrial membrane potential align with the presence of Cyt-c in the cytoplasm [87]. This series of events highlights the emergence of cytoplasmic Cyt-c prior to the initiation of apoptosis in the degradation of the *Drosophila* salivary gland. It sheds light on an aspect of the conserved apoptotic process shared between insects and mammals. In a captivating exploration of the genetic and hormonal orchestration of neuronal death within the *Drosophila* central nervous system, it has been clarified that apoptosis is triggered via the genes’ expression of *reaper*, *grim*, or *head involution defective*. This neuronal demise is intrinsically linked to the decline in the titer of the steroid hormone 20E, a phenomenon characteristic of the metamorphic culmination. As previously mentioned, the pivotal role of *E93* in upregulating downstream secondary response genes has been further underscored. This significance becomes particularly evident when its RNAi knockdown or mutation leads to the inhibition of both autophagy and caspase activity. Conversely, overexpressing *E93* not only activates both processes but also more effectively counters the inhibition of autophagy compared to caspase activity under *EcR* overexpression. *E93* inhibits PI3K-TORC1 signaling, thereby initiating autophagy [54]. In a subsequent investigation, it was observed that the removal of the tissue-specific transcription factor *Fkh* (Fork head) not only triggers a death response to 20E within the larval salivary glands, but it is also sufficient for this initiation. Intriguingly, this loss of *Fkh* is identified as a steroid-regulated event, orchestrated by the 20E-induced *BR-C* gene. Consequently, this sensitizes the pivotal death regulators, *Hid* (head involution defective) and *reaper* to hormonal signals. These findings accentuate the multifaceted role of the *D. melanogaster* FOXA orthologue *Fkh*, positioning it in a previously uncharted context as a crucial competence factor in the domain of steroid-governed cellular demise [89]. Furthermore, 20E inhibits the PI3K/mTOR signaling to activate the ATG1/ATG13 complex, initiating autophagosome formation in the *Drosophila* fat body [90].

Considering the multifaceted functions of steroid hormone 20E signaling, a more comprehensive exploration of its physiological roles and precise mechanisms in autophagy is warranted.

### 4.2. Regulation in B. mori

The intricate interplay between 20E signaling and autophagy in insects, particularly in *Bombyx*, holds great scientific importance. 20E-induced autophagy has been observed during the larval–pupal transition in various insects, with particular emphasis on *Bombyx*. It is noteworthy that 20E is both necessary and sufficient to induce autophagy in larval tissue, accomplished by upregulating the expression of several *Atgs* (autophagy-related genes), implying E-induced autophagy is a vital process in the degradation of larval tissues [91]. In *Bombyx* Bm-12 cells, both 20E treatment and starvation induced cell death, where autophagy preceded apoptosis. Upon exposure to 20E or starvation, BmATG8 underwent rapid cleavage, conjugating with PE (phosphatidylethanolamine) to form BmATG8-PE, followed by the cleavage of BmATG5 and BmATG6 into BmATG5-tN and BmATG6-C, respectively [2]. Studies have revealed that 20E treatment enhanced the transcription of *BmV-ATPase* and this effect was diminished by RNAi targeting the 20E receptor *BmUsp*. The hormone precursor 20E upregulated the transcription of *BmV-ATPases* by inducing the *Bombyx transcription factor EB* and facilitating its nuclear translocation. Additionally, 20E suppressed mTOR signaling, thereby promoting the transcription and assembly of BmV-ATPase subunits [92]. Notably, injecting 20E markedly induced apoptosis and elevated the gene expression of apoptotic within 6 h. When larval fat body tissues were exposed to 20E in vitro, a substantial upregulation was observed in a suite of eight genes integral to the apoptotic pathway. These genes, which include *Apaf-1*, *Nedd2 like1*, *Nedd2 like2*, *ICE1*, *ICE3*, *ICE5*, *Arp*, and *IAP* (inhibitor of apoptosis), showed heightened expression levels during critical periods of molting and pupation. This pattern of gene expression underscores the significant role of 20E in enhancing apoptotic pathways, thereby promoting PCD within the *Bombyx* fat body throughout the larval molting process and the transition into pupation [93]. More importantly, the 20E signaling pathway has been identified to enhance autophagy, primarily via the induction of the *Atg* expression and the simultaneous inhibition of the mTOR pathway. Intriguingly, early-response transcription factors to 20E, including *Br-C*, *E74*, *E75*, and *E93*, play pivotal roles in regulating this autophagy process [94]. Specifically, *Bombyx E75* has been demonstrated to be essential for autophagy induction during the larval–pupal metamorphic transition [55]. The primary response gene to 20E, *E93*, functions through GAGA-containing motifs. It coordinates with the 20E-EcR-USP complex, playing a crucial role in the remodeling of larval tissues and the formation of adult tissues during the *Bombyx* larval–pupal metamorphosis [56]. Research has revealed that the expression of the *E93* gene in mosquitoes is suppressed by JH and stimulated by 20E. This ecdysone-induced protein, E93, plays a crucial role in overseeing the gonadotrophic cycles of adult female mosquitoes, particularly in *Aedes aegypti*. It is worth noting that when *E93* RNAi was applied to silence the gene prior to the initiation of the first gonadotrophic cycle, it led to a disruption in the subsequent cycle’s normal progression [95]. This result indicates that 20E can exert diverse effects on the regulation of *E93* in different species. Furthermore, 20E has been shown to simulate a starvation-like state in *Bombyx*, primarily by diminishing food intake, thereby inducing autophagy [94]. Previous research has emphasized that BmATG5 and BmATG6 undergo cleavage into BmATG5-tN and BmATG6-C, respectively. These cleaved forms play a pivotal role in mediating the transition from autophagy to apoptosis. Notably, this shift is prompted by common stimuli, specifically 20E and starvation [2,96]. Subtle variations in *BmAtg13* have been intricately associated with autophagy, activated either by 20E or via starvation. Interestingly, both the knockdown and overexpression of *BmAtg13* inhibit autophagy. While the 20E treatment significantly elevates *BmAtg13* gene expression, blocking 20E signaling transduction—specifically via the knockdown of *BmUsp*—leads to a reduction in both the gene expression and protein levels of BmAtg13. This emphasizes the pivotal role of *BmAtg13* in 20E- and starvation-induced autophagy in *B. mori*, paving the way for more comprehensive studies [97]. It was also discovered that among the *Atg* genes, five exhibited primary responsiveness to 20E. This was highlighted by the identification of a distinct 20E response element within the *Atg1* promoter region, an ortholog of human *ULK1*. The complexity of this dynamic was further unveiled when the RNAi-mediated knockdown of four key genes—*Br-C*, *E74*, *HR3*, and *βftz-F1*, all integral to the 20E-induced transcriptional cascade—resulted in diminished autophagy and the varied downregulation of *Atg* genes. From these findings, it was deduced that 20E’s influence is not confined to merely inhibiting TORC1 activity for autophagosome creation; it significantly boosts *Atg*’s genes expression, thereby promoting autophagy in the *Bombyx* fat body [94]. However, the exact underlying mechanism remains to be thoroughly elucidated. Furthermore, our findings illuminate that 20E adeptly dephosphorylates the histone deacetylase BmRpd3, orchestrating the deacetylation modification of ATG proteins, culminating in its nucleo-cytoplasmic translocation and the subsequent amplification of autophagy [98]. The potential parallels drawn from 20E signaling in insects may offer foundational knowledge applicable to other organisms, including vertebrates. lncRNAs (Long non-coding RNAs) have been implicated in the regulation of autophagy [99]. A recent investigation identified LNC_000560 and its putative target gene as being associated with 20E-regulated autophagy in *B. mori* [100]. These investigations highlight a profound connection between 20E and the regulation of both autophagy and apoptosis.

In essence, 20E signaling augments autophagy via various pathways, encompassing the induction of *Atg* and *V-ATPases* gene expression, the inhibition of mTOR pathway, and deacetylation modifications of ATG proteins.

### 4.3. Regulation in H. armigera

Beyond the induction of *Atg* gene transcription and mTOR signaling inhibition, 20E’s involvement in autophagy regulation extends even further. Its role in regulating autophagosome formation has been established and research has additionally elucidated the mechanism by which the regulation of ATG12-ATG5 conjugation influences autophagy. Within the context of *H. armigera*, 20E adeptly orchestrates the ATG12–ATG5 conjugation in a manner that is both concentration and time sensitive, thereby promoting autophagy and apoptosis in the insect midgut, a phenomenon that will be elaborated upon subsequently [78]. In a detailed study, it was found that during lepidopteran metamorphosis, the roles of CTSD (cathepsin D), a lysosomal aspartic protease, in apoptosis and cell proliferation are intricately regulated. This is achieved via its varied tissue-specific expression and the precise autophagy-mediated maturation process. Specifically, CTSD boosts cell proliferation by releasing its pro-enzyme as an external ligand, while its mature intracellular form encourages apoptosis. This balance is further emphasized by the 20E, which increases *CTSD* expression, and the orchestrated autophagy process that facilitates its maturation, ultimately culminating in the activation of caspase 3 and the promotion of apoptosis [101]. Recent findings have underscored the central role of 20E in reshaping amino acid metabolism. A novel study indicates that during metamorphic development, 20E intricately orchestrates amino acid metabolism. Notably, arginine, α-KG (alpha-ketoglutarate), and Glu (glutamate) are identified as signature metabolites for feeding larvae, migrating larvae, and pupae stages, respectively. This metamorphic shift involves a 20E-mediated reduction in arginine levels, achieved by inhibiting arginosuccinate synthetase and amplifying arginase expression. In the larval midgut, Glu transforms into α-KG via GDH (glutamate dehydrogenase), a conversion restrained by 20E. Fascinatingly, in the pupal fat body, the process is reversed: α-KG is reverted to Glu by GDH-like, an enzyme reinforced by 20E. These findings underscore 20E’s significant influence in the metabolic reconfiguration, ensuring a seamless insect metamorphosis [6]. While the steroid hormone 20E is known to increase the expression of calcium release-activated *calcium channel modulator 1*, thereby inducing apoptosis in the midgut of *H. armigera*, a contrasting study offers a more nuanced perspective, illustrating that insulin and 20E are involved in a delicate antagonism in modulating *PDK1* (phosphoinositide-dependent kinase-1) expression during insect pupation. This delicate balance is evident, as the dsRNA-mediated *PDK1* knockdown in *H. armigera* larvae not only postpones pupation, resulting in diminutive pupae, but also attenuates Akt/protein kinase B expression while enhancing *FoxO* (Forkhead Box O) expression, suggesting that the *PDK1* knockdown impedes midgut remodeling and reduces 20E levels in the larvae, a phenomenon consistent with the shared regulatory mechanisms of autophagy and apoptosis [102,103]. Probing further into the metabolic complexities of holometabolous insect development, a subsequent study suggests a synchronized interplay between glycolysis and autophagy. This balance is orchestrated via the intricate interrelation of the 20E and insulin signaling pathways. Specifically, insulin accelerates glycolysis and cell proliferation by phosphorylating PGK1. In contrast, 20E employs PTEN to dephosphorylate PGK1, effectively inhibiting glycolysis. This discovery sheds additional light on the subtle role of 20E in regulating autophagy [80]. In the *H. armigera* epidermal cell line, STIM1 (stromal interacting molecule 1) regulates store-operated calcium entry, with 20E-induced phosphorylation at Ser-485 via PKC (protein kinase C) playing a crucial role in calcium ion influx, STIM1 clustering, and its interaction with Orai1 and subsequent induction of apoptosis, further confirming the role of 20E in regulating apoptosis [104]. In contrast, the activation of the *EcRB1/USP1* transcription complex by 20E led to an upregulation of *PKCδ* expression in various tissues during metamorphosis, while silencing *PKCδ* hindered the larval–pupal transition, inhibited tissues from undergoing autophay and apoptosis, and resulted in a decreased expression of the transcription factor *Brz-7*, as well as the apoptotic executors caspase-3 and caspase-6 [105]. This mechanism strongly suggests that the phosphorylation of PKCδ at the threonine residue located at position 1343 was critical for its proapoptotic activity. While 20E plays different roles in different species, leading to diverse impacts, the results of its actions are closely interlinked. Particularly noteworthy is the upregulation of prodeath serine/threonine protein kinase expression for PCD by the steroid hormone 20E, with studies illuminating that in the lepidopteran insect *H. armigera*, the knockdown of prodeath *S/TK*—achieved by injecting dsRNA into the larval hemocoel—not only prevents 20E-induced metamorphosis and PCD, but also attenuates the expression of a number of genes integral to the PCD and 20E signaling pathway, and intriguingly, 20E amplifies prodeath *S/TK* expression via its nuclear receptors *EcR-B1* and *USP1*, further solidifying the pivotal role that 20E plays in autophagy and apoptosis [106].

### 4.4. Regulation in the Others

While *D. melanogaster* exhibits a notably specialized developmental trajectory, suggesting that its apoptotic regulatory mechanisms may not be emblematic of the broader insect class, insights from *B. germanica* (*Blattella germanica*) offer a contrasting perspective. This species is known to possess two inhibitors of apoptosis proteins, with BgIAP1 being indispensable for maintaining tissue vitality, particularly in the prothoracic gland, throughout nymphal maturation. A closer look reveals that the orchestrated degeneration of the prothoracic gland is governed by an intricate 20E-initiated cascade of nuclear receptors, culminating in the potent activation of the death-inducing *Fushi tarazu factor 1* as the nymph transitions to adulthood. Complementing this understanding, it is noteworthy that the JH acts as a formidable safeguard, effectively preventing the degeneration of the prothoracic gland [107].

In the honeybee, *Apis mellifera Ligustica*, a study has shown that 20E triggers apoptosis in the larval fat body during pupation, a process that is mitigated via the RNAi-mediated knockdown of *ECR* genes. The peak expression of the *ECR* gene occurs in seven-day-old larvae, followed by a gradual decline during the pupal stage, highlighting 20E’s decisive role in the regulation of pupation via apoptosis [108]. In *Heortia vitessoides Moore*, research has uncovered expression patterns of *HvATG8*, with a notable upregulation in the prepupal, pupal, and adult stages, as well as in the larval midgut and adult abdomen. Crucially, this suppression of *HvATG8* led to a simultaneous decrease in *HvATG3*, revealing a functional interplay between these genes. Additionally, *HvATG8* expression demonstrated responsiveness to 20E, as well as environmental stressors like starvation and extreme temperatures, indicating its pivotal role in stress response and developmental regulation [109]. During the pupal stage, the orchestrated degeneration of larval cells occurs via autophagic cell death, a process marked by caspase-3 activity and regulated by 20E via the ecdysone receptor *EcR-B1* pathway. This is further substantiated via the increased activity of acid phosphatase and the simultaneous upregulation of *ATG6* and *ATG8*, which align with autophagosome formation in the fat body of *Galleria*, underscoring autophagy’s pivotal role in developmental remodeling [110].

Taken together, the regulation of autophagy and apoptosis by 20E represents a critical point in the cellular physiology of the insects (Table 2). These processes, fundamental to cellular homeostasis and development, are intricately modulated by 20E to ensure that cells are either rejuvenated via autophagy or undergo programmed death via apoptosis at the appropriate developmental stages.

**Table 2 insects-14-00871-t002:** 20E regulated autophagy and apoptosis in insects.

PCD-Related Enzymes	Species	Autophagy/Apoptosis	Up/Downregulation	Pathway	Associated Protein/Factors
Caspase	*D. melanogaster*	Apoptosis	Up [54]	PI3K-TORC1	Caspase-9, Caspase 3, Caspase-6
AMPK-PP2A axis	*D. melanogaster*	Autophagy	Down [15]	Insulin/IGF	PP2A, AKT
ATG1/ATG13 complex	*D. melanogaster*	Autophagy	Down [90]	PI3K/mTOR	PTEN
BmADK	*B. mori*	Autophagy and apoptosis	Down [61]	\	ATG-8, Caspase-9, Ec-R, *E74A*, *Br-C*
BmV-ATPase	*B. mori*	Autophagy and apoptosis	Up [92]	mTOR	TFEB, *BmATG8*, *BmATG5*, *BmATG6*
20E-EcR-USP complex	*B. mori*	Autophagy and apoptosis	Down [94]	BECN1/ATG6-PIK3C3/Vps34 (catalytic subunit of class III PtdIns3K)	*Br-C*, *E74*, *E75*, *E93*
BIGFLP	*B. mori*	Autophagy	Up [60]	Insulin/IGF	Insulin-like peptides
HDACs	*B. mori*	Autophagy	Up [98]	BmRpd3/HsHDAC1	mTORC1
HaP60	*H. armigera*	Apoptosis	Up [62]	Insulin/IGF	ILPs, HaFOXO
PDK1 and FoxO	*H. armigera*	Apoptosis	Up [103]	\	Insulin, calcium channel modulator 1
CTSD/CathD/CATD	*H. armigera*	Apoptosis	Up [101]	Activation of MAPK1/ERK2 -MAPK3/ERK1 and AKT/protein kinase B	Ccaspase 3 and Caspase 7
PGK1	*H. armigera*	Autophagy	Up [80]	\	ARD1, PTEN
PKCδ	*H. armigera*	Apoptosis	Up [105]	EcRB1/USP1	Brz-7, Caspase-3, Caspase-6

## 5. Discussion

In the realm of insect development and metamorphosis, the intricate interplay between 20E and PCD presents a maze of unresolved questions. The role of 20E, as a primary orchestrator of insect growth, raises fundamental questions about its ability to modulate PCD in different tissues and developmental stages. Initially, insect metamorphosis is a remarkable process characterized by extensive PCD, a phenomenon which might not be as pervasive in other organisms. The intricacy of insect metamorphosis necessitates this cell death, ensuring redundant tissues are eliminated to facilitate the emergence of a mature individual. Moreover, the hormone 20E in insects serves as a vital regulator of this PCD. It acts as a guide, directing cells towards their demise by modulating gene expression and internal biochemical processes. Such a regulatory mechanism, and the prominence of this hormone, might not be mirrored or hold equivalent significance in other organisms. Subsequently, the occurrence of PCD in insects is intricately timed, coinciding with specific developmental stages and tissues. This is particularly evident during phases like metamorphosis and tissue remodeling. Contrastingly, the patterns of cell death in other organisms may diverge, given that different species have their own unique developmental trajectories and life cycle demands. Collectively, while there might be overarching similarities in the regulation of PCD across various organisms, insects exhibit unique characteristics and mechanisms. These are intrinsically tied to their biological attributes and developmental paradigms. One wonders whether the cellular response to 20E is a direct, immediate one, or whether it is a more complicated process that unfolds in a series of nuanced steps, revealing a complex choreography of cellular events. Beyond the well-established EnR pathway, there is speculation about potential alternative pathways—hidden alleys and byways, through which 20E might exert its influence on PCD.

The emergence of pesticide-resistant insect strains raises another pressing question: To what extent are alterations in 20E signaling or perturbations in PCD pathways responsible for this burgeoning resistance? Such resistance not only poses challenges for agriculture and pest management, but also underscores the need to understand the molecular and physiological intricacies that underlie these changes.

From an ecological perspective, the implications of 20E-induced PCD go beyond the individual insect. What are the cascading effects on ecosystems when 20E pathways are altered, either naturally or via human intervention? The potential impacts on food webs, species interactions, and ecosystem stability warrant thorough investigation. Finally, the evolutionary perspective offers a rich tapestry of insights. The 20E-PCD relationship, which has persisted across evolutionary timescales, prompts reflection on its significance in the insect evolutionary narrative. Does this relationship represent a conserved developmental mechanism that has been essential for insect success? Or does it suggest a dynamic interplay, constantly reshaped by ecological pressures and evolutionary forces? Beyond its established roles in autophagy and apoptosis, how might 20E influence other forms of PCD, such as necroptosis or pyroptosis? How does the regulation of PCD in insects compare to that in other organisms? Could 20E also play a pivotal role in the insect reproductive processes? If so, what are the underlying mechanisms through which these hormones operate? Are there specific insect species or developmental stags where 20E’s influence on PCD mechanisms other than autophagy and apoptosis is more pronounced? Combining these questions, it becomes evident that the 20E-PCD axis in insects transcends being a mere biological mechanism; rather, it represents a multifaceted subject intertwining ecology, evolution, and applied science. Addressing these questions is not only pivotal for advancing our understanding of insect biology, but also for addressing the challenges and opportunities they present in a rapidly changing world.

In summary, a comprehensive understanding of the 20E signaling landscape, highlighting its important role in mediating autophagy and apoptosis in insects and clearly revealing the mechanisms behind this action, will provide new avenues in our research (Figure 2) [111]. By delving into the 20E-autophagy/apoptosis axis, we gain profound insights not only into the complex interconnections within endocrine networks, but also into the innovative adaptive strategies insects have evolved in response to environmental challenges.

## Figures and Tables

**Figure 1 insects-14-00871-f001:**
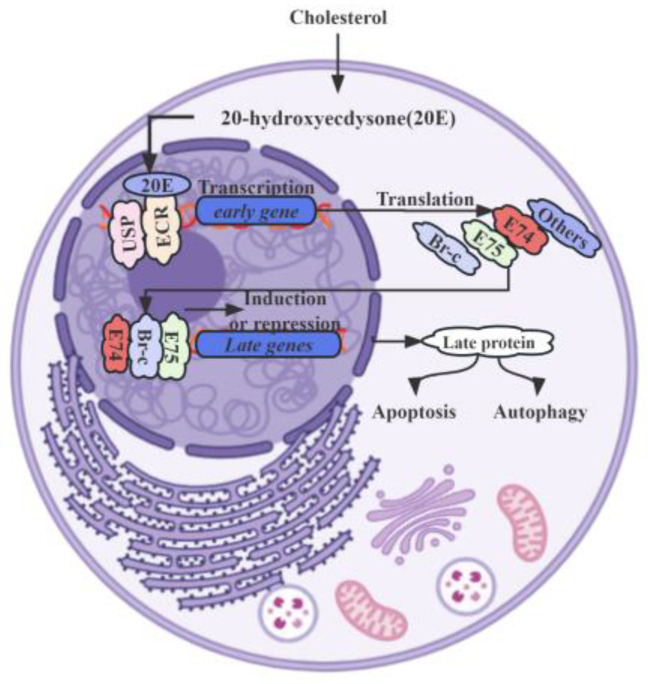
EcR-USP swiftly and potently induces the expression of primary-response genes upon 20E stimulation.

**Figure 2 insects-14-00871-f002:**
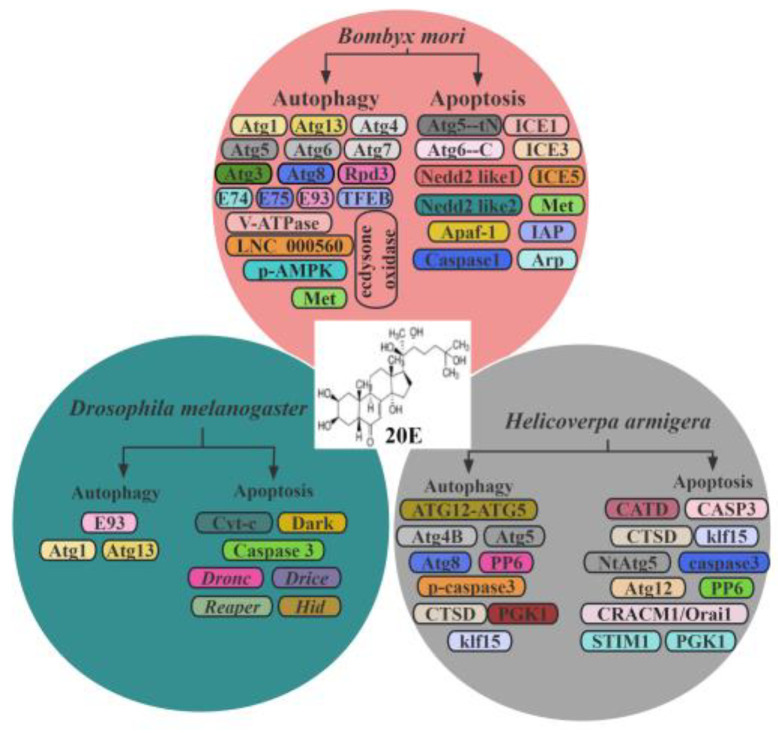
20E regulated autophagy and apoptosis in insects.

## Data Availability

Not applicable.

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
