# Peer review of "Research Progress on the Regulation of Autophagy and Apoptosis in Insects by Sterol Hormone 20-Hydroxyecdysone"

_insects, 2023, doi:10.3390/insects14110871_

Round 1

Reviewer 1 Report

Comments and Suggestions for Authors

In this manuscript, Lin et al summarize recent research on the programmed cell death in insects by 20E and discuss the molecular mechanisms by which 20E regulates these processes, as well as the role of 20E in driving developmental changes and metamorphosis in arthropods. In addition, the writing needs to be carefully checked and polished to fix the language issue. Comments are listed below:

1.       Please confirm the accuracy and grammar of words, such as line 18, 19, 31 and 65 etc.

2.       What is the role of the hormone 20-hydroxyecdysone in regulating programmed cell death in insects?

3.       How does the regulation of programmed cell death in insects compare to that in other organisms?

4.       Finally, while the article provides a useful overview of the current state of research in this field, it does not offer any definitive conclusions or recommendations for the future research direction.

5.       One of important paper (Wang et al, 2021 PNAS e2021910118) about 20E induced E93 governing the fat body autophagy in mosquitoes should be cited.

6.       Figures are obscure, specially 20E molecular formula in Fig 2.

Comments on the Quality of English Language

There are lots of small grammar errors detected.

Author Response

Dear Reviewer,

Thank you for taking the time to review our manuscript and for your valuable feedback. We greatly appreciate your detailed comments and suggestions. Below, please find our responses to your comments.

Warm regards,

Luobin Lin and Wenmei Wu

Reviewer 2 Report

Comments and Suggestions for Authors

The manuscript, entitled Research Progress on the Regulation of Autophagy and Apoptosis in Insects by Sterol Hormone 20-Hydroxyecdysone, presented by Lin and co-authors, is devoted to an interesting and topical issue: the hormonal regulation of cell death in insects. This topic has both applied (agriculture) and fundamental significance. It's good that the authors emphasize this in their review. The authors wrote the abstract well and intriguingly, so after reading the abstract, there was a desire to read the review. However, the text of the review is written carelessly. It is difficult to read and sometimes impossible to understand.

1.Some sentences seem absurd. For example, “The primary active form of the insect steroid hormone ecdysone regulates vitellogenin synthesis in the fat body after a blood meal [7], thereby, resulting in titer fluctuations that helps regulate insect development, reproduction, which promotes nucleic acid, protein synthesis, glucose metabolism, and lipid metabolism in various tissues and organs of the holometabolous insects [4].” It is not clear what the authors want to say. They're probably it is about blood-sucking insects, perhaps mosquitoes? Then why do the authors refer to [7], Yuan et al., “The AMPK-PP2A axis in insect fat body…”? This article does not contain information about “vitellogenin synthesis in the fat body after a blood meal”. It is also unclear what  are «titer fluctuations»? 

One more example, “It was observed that in vitro exposure of larval fat body tissues to 20E led to the upregulation of all eight apoptotic genes, which underscores 20E's role in enhancing apoptotic gene expression and inducing apoptosis in the Bombyx fat body during larval molting and the larval-pupal transition [82]”.  It seems that the authors believe that there are only 8 apoptotic genes, and these genes are generally known, so the authors do not specify them.

Unfortunately, through the text, there are a lot of such strange sentences. The authors try to provide a large amount of data, but they do not demonstrate systematization and deep analysis of these data, and, unfortunately, they formulate their conclusions poorly.

2. Another major problem with this review is the lack of clear definitions. Specifically, the authors use the term “autophagy,” but it is unclear whether they mean cell death through autophagy or a process that promotes cell survival. Therefore, confusion arises; it is unclear what mechanism 20E is involved in.

Through the text, authors sometimes write ecdysone rather than 20-Hydroxyecdysone. How correct is this? I understand that ecdysone is a precursor to 20E, but this hormone has other functions as well.

I believe that at the beginning of the review (Introduction) it is necessary to provide clear definitions and a brief description of processes such as autophagy and apoptosis, focusing on those aspects that are discussed in the manuscript. It is also necessary to clarify the connection and differences between ecdysone and 20E at the very beginning of the review.

3. The lack of clear definitions, poorly formulated sentences and phrases, and sometimes incorrect citing and references make the text difficult to read. Authors need to better comprehend the information and rewrite the text, taking care of clarity of wording and connections between sentences. It is also necessary to carefully check the correctness of the citation.

Author Response

Dear Reviewer,

We deeply appreciate your insightful feedback. Your suggestions are not only perceptive and clear but also embody the kind of constructive critique we highly value. After thoughtful deliberation, we wholeheartedly concur with your observations and have made revisions in accordance with your recommendations. In our forthcoming responses, we will attentively address each of your points. This exchange has been instrumental in enhancing the quality of our manuscript, and we look forward to further discussions that will further enrich our work. All revisions are highlighted in blue.

Warm regards,

Luobin Lin and Wenmei Wu

Reviewer 3 Report

Comments and Suggestions for Authors

A review paper by Lin et al. summarized recent advances in the regulation of autophagy and apoptosis by ecdysteroids. This topic has attracted the attention of many researchers, especially in the research fields of developmental biology and reproductive biology. In general, excellent review papers should provide not only basic knowledge, recent findings and previously proposed hypotheses regarding this topic, but also new perspectives, remaining problems and new hypothesis that solve the problem. This paper provided basic knowledge and recent findings, but did not introduce any hypotheses, current problems and authors’ suggestions. Therefore, this paper appears to have little originality. Please provide your own insights and hypothesis for current issues.

Regarding the reproductive roles by ecdysone (line 43-53), the role has been confirmed in a limited number of taxa including Diptera. However, other groups such as Coleoptera and Hymenoptera use juvenile hormone primarily for vitellogenesis in the fat body. Therefore, reproductive function by ecdysone is a minority among holometabolous insects. The descriptions in this paper may be misleading to readers and should be revised to take into account other taxa. The following papers may be helpful.

Raikhel, A. S., Brown, M. R., Belles, X. (2005) Hormonal control of reproductive process. In: Comprehensive Molecular Insect Science, Eds. By Gilbert, L. I, Latrou, K. and Gill, S. S., Chapter 3.9. p433-491. Elsevier.

There were careless mistakes in spelling, section titles and italics, etc. Gene names should be written in italics throughout the text and Table 1. There is no citation for Figure 2 in the text.

Figure 1: This figure shows that translated proteins (E74, E75 and Br-c) induce or repress expression of late genes in the cytoplasm. Why is double-stranded DNA present in the cytoplasm?

Section 4 (lines 234-454): This section is too long and should be shortened. It would be better to separate “Drosophila melanogaster” (lines 263-311), “Bombyx mori” (lines 312-377) and “Helicoverpa armigera” (lines 378-436) as subsections with subheadings.

The authors should also make effective use of figure 2. For example, genes that overlap between species can be shown in the same color.

Several review papers nicely summarize recent findings and are useful for presentation. These should be cited in the text.

Nicolson, S., Denton, D., Kumar, S. (2015) Ecdysone-mediated programmed cell death in Drosophila. Int. J. Dev. Biol. 59, 23-32.

Tettamanti, G., Casartelli, M. (2019) Cell death during complete metamorphosis. Philos. Trans. R. Soc. Lond. B 374, 20190065.

Comments on the Quality of English Language

There were careless mistakes in spelling, section titles and italics, etc.  

Author Response

Dear Reviewer,

We are immensely grateful for your insightful suggestions. They have proven to be tremendously beneficial, expanding our perspective and enriching the content and depth of our article. Your dedicated time and effort in conducting this review are deeply appreciated. We will thoroughly address each of the points you've raised. This exchange has had a profound impact on us, and we truly value your guidance. All revisions are highlighted in blue.

Warm regards,

Luobin Lin and Wenmei Wu

Round 2

Reviewer 1 Report

Comments and Suggestions for Authors

The quality of this manuscript was improved after revision. Now, it could be accepted.

Author Response

Dear reviewer,

I appreciate your thoughtful review of our manuscript and the valuable feedback you've provided. Your professional insights play a pivotal role in advancing our research, and we are truly grateful for your dedicated effort amidst your busy schedule.

Have a nice day.

Warm regards,

Luobin Lin and Wenmei Wu

Guangdong Province Key Laboratory of Biotechnology Drug Candidate

School of Life Sciences and Biopharmaceuticals

Guangdong Pharmaceutical University

Guangzhou, China 51000

Reviewer 2 Report

Comments and Suggestions for Authors

The authors have submitted a revised version of the manuscript and I agree that it is improved. However, some problems remain. Reviewis devoted to the systematization and analysis of data, obtained as a result of studying the role of the steroid hormone 20-hydroxyecdysone (20E) in the regulation of programmed cell death (PCD) during development, in particular, the metamorphosis of insects. However, authors often deviate from the purpose of the review and describe in detail other physiological functions of the 20E or ecdysone, or discuss various aspects of cell death that are not related to development. Sometimes authors suddenly change contexts and because of this the main idea is lost. This is especially true for the first chapters of the article.  In particular, in the introduction (54–67), they suddenly switch from characterizing 20E to description of juvenile hormone and its role in vitellogenesis. How does this relate to the main topic of the review?

The authors then write: «Signal transduction of 20E is mediated through its binding to specific nuclear receptors, primarily the EnR (ecdysone receptor) [13]. Upon 20E binding, EcR forms a heterodimer with the USP (ultraspiracle protein), leading to the activation of target genes responsible for various developmental processes [14]. Meanwhile, while this enzyme activity impacts both rat and human liver microsomes, delving into the signal transduction of P450 enzymes unveils the depth of 20E's involvement in autophagy and related regulatory processes [15].» The first and second sentences have lost connection with the third. Which enzyme are the authors referring to, P450? Why do they write about “rat and human liver microsomes”? Neither humans nor rats have a molting hormone.

The authors go on to describe 20E Biosynthesis (they think it's important), but it looks confusing and incomprehensible. I think the authors should start with the conversion of cholesterol to ecdysone in the prothoracic glands in larvae and pupae….. (171-181), and then present the regulation of this process.

The paragraph 20E Signal Transduction is confusingly written and  difficult to understand. I think the main problem here is that the authors intertwine data obtained on larvae/pupae and adult flies. In larvae and pupae, 20E induces PCD during metamorphosis. In adult flies it plays a gonadotropic role. These are different mechanisms. Therefore, data must be presented taking these features into account.

Section of the article “20E Regulates Autophagy and Apoptosis in Insects” has improved in the new version, but there are still “strange” sentences.

369-370: “When larval fat body tissues were exposed to 20E in vitro, all eight genes  associated with apoptosis.”

363-365: “Taken together, the cytochrome P450 enzymes and primary-response genes in 20E 263 biosynthesis in different species (Table 1). Essentially, the precision with which 20E signaling modulates gene expression and cellular activities emphasizes its evolutionary importance and central role in insect biology”.

Such unfortunate formulations are often found in the text. I recommend giving the text to an English-speaking colleague to read so that he can help correct the defects.

At the moment, I believe that the manuscript is still not suitable for publication and needs extensive revisions. It is quite voluminous and overflowing with information. The authors should try to make it more logical and easy to understand.

Author Response

Dear reviewer,

Thank you for taking the time out of your busy schedule to review our manuscript and provide valuable feedback. Your professional insights are crucial in enhancing our research, and we deeply appreciate your hard work.

Under your guidance, we have carefully considered all of the suggestions you provided and made corresponding revisions. Below are some significant changes we have implemented based on your advice.

Warm regards,

Luobin Lin and Wenmei Wu

Guangdong Province Key Laboratory of Biotechnology Drug Candidate

School of Life Sciences and Biopharmaceuticals

Guangdong Pharmaceutical University

Guangzhou, China 51000

Reviewer 3 Report

Comments and Suggestions for Authors

The Manuscript has been significantly improved from the previous version. The content of the manuscript is sutable for a special issue of this journal.

Author Response

Dear Reviewer,

I am immensely grateful for the time and effort you dedicated to reviewing our manuscript. Your perceptive comments and suggestions have played an indispensable role in elevating the caliber of our research. Your expertise and unwavering commitment to the peer review process are deeply valued. Once again, thank you for your invaluable contribution.

Warm regards,

Luobin Lin and Wenmei Wu

Guangdong Province Key Laboratory of Biotechnology Drug Candidate

School of Life Sciences and Biopharmaceuticals

Guangdong Pharmaceutical University

Guangzhou, China 51000

Round 3

Reviewer 2 Report

Comments and Suggestions for Authors

The manuscript has been significantly improved, taking into account my  remarks